# Density Functional Theory Studies on the Chemical Reactivity of Allyl Mercaptan and Its Derivatives

**DOI:** 10.3390/molecules29030668

**Published:** 2024-01-31

**Authors:** Marcin Molski

**Affiliations:** Department of Quantum Chemistry, Adam Mickiewicz University of Poznań, ul. Uniwersytetu Poznańskiego 8, 61-614 Poznań, Poland; mamolski@amu.edu.pl

**Keywords:** garlic metabolites, 2-propenesulfenic acid, allyl mercaptan, thermodynamic descriptors, chemical activity descriptors, DFT method

## Abstract

On the basis of density functional theory (DFT) at the B3LYP/cc-pVQZ level with the C-PCM solvation model, a comparative analysis of the reactivity of the garlic metabolites 2-propenesulfenic acid (PSA) and allyl mercaptan (AM, 2-propene-1-thiol) was performed. In particular, the thermodynamic descriptors (BDE, PA, ETE, AIP, PDE, and G_acidity_) and global descriptors of chemical activity (ionization potential (IP), electron affinity (EA), chemical potential (μ), absolute electronegativity (χ), molecular hardness (η) and softness (S), electrophilicity index (ω), electro-donating (ω^−^) and electro-accepting (ω^+^) powers, and Ra and Rd indexes) were determined. The calculations revealed that PSA is more reactive than AM, but the latter may play a crucial role in the deactivation of free radicals due to its greater chemical stability and longer lifetime. The presence of a double bond in AM enables its polymerization, preserving the antiradical activity of the S-H group. This activity can be amplified by aryl-substituent-containing hydroxyl groups. The results of the calculations for the simplest phenol–AM derivative indicate that both the O-H and S-H moieties show greater antiradical activity in a vacuum and aqueous medium than the parent molecules. The results obtained prove that AM and its derivatives can be used not only as flavoring food additives but also as potent radical scavengers, protecting food, supplements, cosmetics, and drug ingredients from physicochemical decomposition caused by exogenous radicals.

## 1. Introduction

The extracts and isolated chemical constituents of garlic, *Allium sativum*, are endowed with various bioactive properties, including antibacterial, antifungal, antiprotozoal, antiviral, anti-inflammatory, antiradical, anticancer, antimutagenic, anti-asthmatic, anti-amnesic, hepatoprotective, neuroprotective, hypotensive, hypoglycemic, immunomodulatory, urease/xanthine oxidase inhibitory, and prebiotic activity [1]. The various bioactive properties of garlic, particularly its antiradical activity [2,3,4,5], deserve special attention. In particular, the mechanism of the neutralization of reactive free radicals by garlic has not yet been fully recognized. In general, one may indicate two scenarios of free radical scavenging by the active substances of garlic: indirect and direct. In the first case, the antiradical effect is connected with the increasing synthesis of endogenic antioxidants—glutathione and enzymes (catalase, glutathione reductase, glutathione peroxidase, etc.)—or with the decreasing production of oxidizers, such as reactive oxygen species (ROS), through stimulation of the synthesis of nitric oxide (NO) via the activation of NO synthase (eNOS) and the inhibition of NADPH oxidase (Nox), which is responsible for ROS overproduction [4,5,6,7,8]. In the simplest direct scenario, free radical species are scavenged by selected garlic components endowed with antioxidant activity. According to Okada et al. [9], the first candidate for such a compound is allicin (diallyl thiosulfinate), which is formed from alliin (S-allyl-L-cysteine sulfoxide), the main chemical constituent of garlic, compartmentalized in mesophyll cells. Under the influence of alliinase, which is localized in vascular bundle sheath cells and released by the physical disruption (chopping or cutting) of garlic bulb tissues, alliin is converted to PSA [10] (Figure 1).

In the next step, PSA undergoes self-condensation to convert allicin to diallyl disulfide and AM, which are the volatile compounds responsible for the specific odor of crushed garlic and, presumably, its health benefits. It should be pointed out that PSA can be formed not only from alliin but also from allicin, which undergoes Cope elimination, producing PSA and inactive thioacrolein. Okada et al. [9] suggested an antioxidant mechanism involving the abstraction of the allylic hydrogen atom adjacent to the sulfur atom in allicin (Figure 2A). This model has been criticized by Vaidia et al. [10], who argue that the rate constants for hydrogen transfer from hydrocarbons to peroxyl radicals are much lower than those reported for allicin inhibited by autooxidation reactions. Furthermore, carbon-centered radicals react with O_2_, producing new peroxyl radicals, which generate the autoxidation chain reaction. It is also worth mentioning that allicin is not found in the blood after garlic consumption as it reacts rapidly with cysteine to form S-allyl mercaptocysteine and then is converted to AM. Investigations in this regard have revealed [11] that allicin and some of its transformation products are metabolized to AM shortly after entering into the circulation, although they may first be metabolized to S-allyl mercaptocysteine in the intestinal tract. In such circumstances, Vaidia et al. [10] proposed the use of PSA, which is a much stronger scavenger than allicin, as it is the compound responsible for the direct antiradical activity of crushed garlic (Figure 2B). PSA is a highly reactive species and an effective scavenger [12] of peroxyl radicals, ROO^●^. It plays an important role in redox-mediated signal transduction in biological systems [13], and it is connected with the oxidation of cysteine residues in proteins [14]. Unfortunately, Vaidia et al.’s [10] thesis is problematic since PSA is a short-lived [15] species (less than 1 s in the gas phase). Therefore, the interval of its antiradical activity is strongly restricted, which makes it impossible to use PSA as an active component of supplements, cosmetics, or drugs. Due to the high chemical stability of AM, its bioavailability is much greater than that attributed to PSA. AM is one of the volatile organic compounds responsible for what is known as “garlic breath”. The classes of these substances include [16] diallyl disulfide (48.6 mg in 100 g of raw garlic), allyl methyl disulfide (8.16), allyl methyl sulfide (0.37), and AM (0.06). An unpleasant odor that persists even for a few days after garlic consumption is indirect proof of its chemical stability and, consequently, the chemical activity of AM, which lasts longer than PSA. Additionally, the presence of the thiol S-H group guarantees adequate efficiency in radical scavenging similar to that observed, for example, in the glutathione and α-lipoic acid [17] metabolites (bisnorlipoic and tetranorlipoic acids), which are strong endo- and exogenous radical scavengers. In light of the facts mentioned above, the main purpose of the present study is to consider AM as the third candidate responsible for the direct antiradical activity of crushed fresh garlic cloves (Figure 2C). To this aim, a comparative analysis of the reactivity of the garlic metabolites PSA and AM is performed. In particular, the thermodynamic descriptors (BDE, PA, ETE, AIP, PDE, G_acidity_) and global descriptors of chemical activity (ionization potential (IP), electron affinity (EA), chemical potential (μ), absolute electronegativity (χ), molecular hardness (η) and softness (S), electrophilicity index (ω), electro-donating (ω^−^) and electro-accepting (ω^+^) powers, and Ra and Rd indexes) are determined by taking advantage of quantum chemical calculations. We also aim to determine the reactivity parameters for AM derivatives that have stronger antiradical properties than AM and high applicative potential.

## 2. Computation Details

Despite their simple chemical structures and their importance for medical and dietetic applications, PSA and AM have not been intensively studied by means of quantum chemical techniques. In one of the few papers on this subject, the bond dissociation enthalpy BDE = 68.6 [kcal mol^−1^] of the hydroxyl group in PSA and its alkane derivatives in the gas phase were reported by Vaidia et al. [10]. In a more advanced study conducted by Galano et al. [12], the CBS-QB3 quantum method was employed at the BHandHLYP/6-311++G(d,p) level of the theory to explain the antiradical activity of PSA in water. To this aim, three different mechanisms of radical scavenging were considered for the reaction of PSA with the HOO^●^ radical in aqueous solution: radical adduct formation (RAF), hydrogen atom transfer (HAT), and single electron transfer (SET). Calculations revealed that the abstraction of hydrogen from the hydroxyl moiety is the main channel for the reaction, and PSA is over 1000 times more reactive toward the HOO^●^ radical than allicin. These results support the conclusion of Vaidia et al. [10] that the best candidate for the compound responsible for the antiradical properties of garlic is PSA, and not allicin itself. AM has so far not been studied in such a detailed way; therefore, its thermodynamic descriptors and the global parameters of its chemical activity are yet unknown. Motivated by this, in this work, we determine the chemical and thermodynamic characteristics of PSA and AM to compare their ability to scavenge free radicals in a hydrophilic (water) medium, which is a natural environment for their bioactivity. Additionally, we perform calculations for AM in a vacuum and in a hydrophobic medium (benzene) to examine the influence of the medium type on the antiradical activity of AM. An analysis of the descriptors evaluated enables the prediction of the scavenging scenario for the compounds considered. On account of the studies carried out by Vaidya et al. [10], which revealed that hydrocarbon R in the RSOH-type compounds has no or only a marginal effect on their antioxidant potency, the conclusions of the present work can be extended to include a wide range of sulfenic acids and derivatives of AM that do not appear in nature. Since the application of such a poor basis set as 6-311++G(d,p), employed by Galano et al. [12], may result in the imprecise determination of the molecular characteristics, an additional aim of the present study is to perform computations using a more advanced basis set, cc-pVQZ. This approach allows for an accurate evaluation of the descriptors indispensable in characterizing the chemical reactivity, as well as of the preferred mechanism of radical scavenging by the compounds considered. To this purpose, we determined the following: (i)the global descriptors of the chemical activity: ionization potential (IP), electron affinity (EA), chemical potential (μ), absolute electronegativity (χ), molecular hardness (η) and softness (S), electrophilicity index (ω), electro-donating (ω^−^) and electro-accepting (ω^+^) powers, as well as Ra and Rd indexes;(ii)the thermodynamic descriptors: bond dissociation enthalpy (BDE), adiabatic ionization potential (AIP), proton dissociation enthalpy (PDE), proton affinity (PA), electron transfer enthalpy (ETE) and the free Gibbs acidity (G_acidity_).

They are indispensable in characterizing the ability of PSA and AM to scavenge free radicals and chelate transition metal ions (especially Fe^2+^ and Cu^2+^), which may participate in the creation of radicals. The descriptor definitions and exact mathematical formulae used in the calculations are presented in the Appendix A. To calculate the thermodynamic and chemical activity descriptors for the compounds considered, we used the DFT method implemented in the Gaussian vs. 16 software. To this aim, the B3LYP level of the theory, consisting of Becke’s [18] exchange functional in conjunction with the Lee-Yang-Parr [19] (LYP) functional, was taken into account. The input structures were constructed by taking advantage the Gauss View-6.1 graphical interface, whereas the calculations were carried out in the Supercomputing and Networking Center via the PL-Grid Infrastructure. The values of the global and thermodynamic descriptors were calculated using the Maple vs. 16 processor for symbolic computations. In the calculations, we used the following values of the electron, proton, and hydrogen enthalpies in the gas phase (in [Ha] unit): H(e^−^) = 0.001198, H(H^+^) = 0.002363, H(H^●^) = −0.497640; in water: H(e^−^)_aq_ = −0.03879545, H(H^+^)_aq_ = −0.38690958, H(H^●^)_aq_ = −0.49916356; and in benzene: H(e^−^)_be_ = −0.00146823, H(H^+^)_be_ = −0.33815566, H(H^●^)_be_ = −0.495202304 [20]. The last six values can be calculated using the following relationships:H(e^−^)_sol_ = H(e^−^) + ∆_sol_H(e^−^), H(H^+^)_sol_ = H(H^+^) + ∆_sol_H(H^+^), H(H^●^)_sol_ = H(H^●^) + ∆_sol_H(H^●^)
in which (in [kJ mol^−1^] unit): ∆_aq_H(e^−^) = −105, ∆_aq_H(H^+^) = −1022, ∆_aq_H(H^●^) = −4.0, ∆_be_H(e^−^) = −7, ∆_be_H(H^+^) = −894, and ∆_be_H(H^●^) = 6.4 are the solvation corrections recommended by Rimarčik et al. [20]. The geometry optimization and enthalpy calculations were performed using the DFT method at the B3LYP/cc-pVQZ level of theory, including the correlation consistent polarized valence quadruple zeta basis set (cc-pVQZ) introduced by Dunning [21]. The basis set used allows a compromise between the extensive calculation time and the accuracy of the generated results, which may increase for more advanced basis sets, e.g., aug-cc-pVQZ, cc-PV5Z, or cc-PV6Z. However, the test calculations showed that such extensive basis sets do not significantly affect the values of the total energy, the descriptors calculated, or the interpretation of the results obtained. For example, the application of the basis set aug-cc-pVQZ produces a difference in the calculated PSA total energy equal to 1.1 [kcal mol^-1^]. This is the accuracy level required in realistic chemical predictions. In a hydrophilic (water) medium, as well as, in the case of AM, in the gas phase and in a hydrophobic (benzene) environment, the calculations were performed by taking advantage of the Conductor-Like Polarizable Continuum Model (C-PCM) solvation model [22]. The C-PCM model and the test calculations performed for PSA in water revealed that the B3LYP theory level yielded the smallest value of the total energy for this molecule in comparison to the energies generated by other functionals (e.g., M06–2X, ωB97XD, BHandHLYP, CAM-B3LYP). For example (in [Ha] unit), E(M06-2X) = −591.307805, E(ωB97XD) = −591.347034, E(BHandHLYP) = −591.294711, E(CAM-B3LYP) = −591.330316 in comparison to E(B3LYP) = −591.436087. We decided to use the C-PCM model because calculations for gallic acid showed [23] that the B3LYP/C-PCM combination correctly reproduces the difference in HOMO–LUMO energies (∆E). Since these energies determine the values of the chemical activity parameters, the C-PCM model was employed in the calculation of both the chemical activity and the thermodynamic descriptors for the molecules considered. The results of the calculations are graphically displayed in Figure 1, Figure 2, Figure 3, Figure 4 and Appendix A and reported in Table 1, Table 2, Table 3, Table 4 and Table 5.

## 3. Results and Discussion

The thermodynamic descriptors presented in Table 1 reveal that the values of PA = 58 and 50 [kcal mol^−1^] are less than BDE = 68 and 87 [kcal mol^−1^] as well as AIP = 114 and 125 [kcal mol^−1^], respectively. Consequently, the sequential proton loss electron transfer (SPLET) is the dominant free radical scavenging mechanism predicted for PSA and AM in a water medium. Additionally, ETE = 56 and 83 [kcal mol^−1^]; hence, the second step of the SPLET mechanism also requires less energy than that indispensable to activate the HAT or SET-PT processes. The total energy needed to activate both stages of SPLET is represented by PA + ETE = 114 and 133 [kcal mol^−1^] for PSA and AM, respectively. This means that PSA is a stronger antioxidant than AM, which also scavenges free radicals in water through the SPLET mechanism. In a vacuum (benzene), AM deactivates free radicals via the HAT scenario as its BDE = 86 (89) [kcal mol^−1^] is smaller than PA = 355 (108) and AIP = 202 (171) [kcal mol^−1^], respectively. Hence, in a vacuum and in a hydrophobic medium, the SPLET and SET-PT mechanisms will not be activated. The descriptor G_acidity_ = 302 and 320 [kcal mol^−1^], which is related to the TMC mechanism, takes large values for both compounds considered; consequently, metal chelation is not preferred in hydrophilic (water) and hydrophobic (benzene) solutions.

The values of the global descriptors of PSA in water and AM in water, a vacuum, and benzene, presented in Table 2, indicate that the first of the compounds mentioned has ΔE = 5.6461 [eV] (Figure 2), in comparison to 6.3773 [eV] (Figure 3) for the second one; hence, PSA features greater chemical activity than AM. The smaller value of IP = 6.4937 [eV] in comparison to IP = 6.7914 [eV] indicates the greater tendency of PSA to participate in electron transfer relative to AM; hence, the first compound is a stronger radical scavenger than the second one. Important information regarding the chemical activity of the molecules considered is provided by the acceptance (Ra) and donation (Rd) indexes [25]. An inspection of Table 2 reveals that Ra and Rd for PSA (AM) take the values Ra = 0.2657 (0.1859) and Rd = 1.3184 (1.2206). Consequently, both compounds are less effective electron acceptors than the F atom and more effective electron donors than the Na atom. In this regard, PSA has an electro-accepting power identical to that of cyanidin and pelargonidin (anthocyanins) with Ra = 0.27 [25], whereas its AM activity is similar to that of vitamin A with Ra = 0.20 [25], respectively. 

On the other hand, PSA and AM have an electro-donating capacity similar to that of vitamin C, with Rd = 1.29 [25], and delphinidin, with Rd = 1.23 [25], respectively. Additional relevant information regarding the chemical activity of the compounds considered can be deduced from the descriptors characterizing the electro-donating (ω^−^) and electro-accepting (ω^+^) power of a radical scavenger. For PSA (AM), ω^−^ = 4.5746 (4.2353) indicates that the antioxidant activity of PSA is weaker than that assigned [25] to β-carotene, astaxanthin, and psittacofulvins (pigments found in parrot feathers), and stronger than that of vitamins A, C, and E and anthocyanins. The antioxidant power of AM is greater than that reported [25] for vitamins A and E and anthocyanins (except delphinidin), and smaller than that of the remaining cases under comparison [25]. The parameter ω^+^ = 0.9040(0.6326) for PSA (AM), which characterizes the tendency of an anti-reductant to eliminate free radicals by means of electron capture, reveals that PSA is a stronger anti-reductant than vitamins A, C, and E and a weaker one than β-carotene, astaxanthin, psittacofulvins, and anthocyanins. In the case of AM, it is a stronger anti-reductant than vitamins C and E and a weaker one than vitamin A, β-carotene, astaxanthin, psittacofulvins, and anthocyanins. The ability of AM to scavenge free radicals via the HAT mechanism in the gas phase can be compared with the potency of other antioxidants using their BDE (in [kcal mol^−1^]) determined so far. For example, α-lipoic acid enantiomers and their natural metabolites, bisnorlipoic and tetranorlipoic acids, in the oxidized and reduced forms have BDE = 85–87 [17]; trans(cis)-resveratrol and its derivatives, BDE = 69–82 [26]; trans-resveratrol analogs, BDE = 69–83 [27]; trans-resveratrol ethers (arachidins), BDE = 74–77 [28]; trans-p-coumaric acid, BDE = 83 [29]; trans-sinapinic acid, BDE = 77 [29]; and trans-ferulic acid, BDE = 323–330 [30]. Contrary to PSA, which, due to its low stability and short lifetime, cannot be used as an active component of cosmetics, supplements, and drugs, synthetic AM has great applicative potential. The presence of a double bond in AM enables its polymerization (as in the case of propylene), which preserves the antiradical active S-H groups (Figure 5A). The calculated values of BDE in a vacuum for AM dimers and trimers (Table 3) indicate that their antioxidant activity does not decrease (polymer B) or increases slightly (polymer A), which suggests that, in a longer polymer, thiol groups will effectively deactivate the free radicals. However, the well-known continuous-phase glow discharge polymerization of thiol-containing monomers may have been compromised by oxidation. Stynes et al. [31] demonstrated how this technique can be used to polymerize AM to generate thiol groups with significant densities on a mixed polyurethane/tantalum medium. In view of this, the synthesis and practical application (e.g., in food packaging) of such an active polymer seem to be feasible and worthy of attention. The antiradical and antimicrobial properties of AM can also be employed in food processing and storage, e.g., meat preservation through the use of a microfilm polymer containing antioxidant and antimicrobial thiol groups, which would increase the quality of food as well consumer safety. The results obtained for the chitosan/gallic acid system encourage further research in this field [32]. Useful in this regard may be the discovery made by Xi et al. [24] of the new poly(ethylene glycol) and poly(allyl mercaptan) polymers (Figure 5B), which generate an active amphiphilic polymeric coating. This technology displays efficacy both with and without additional active components (e.g., antibiotics). 

Investigations in the field of garlic organosulfur compounds have revealed that AM is an effective histone deacetylase (HDAC) inhibitor, which means that it reactivates epigenetically silent genes in cancer cells and, as a result, initiates cell cycle arrest and apoptosis [33]. This discovery has initiated the in silico modeling of AM derivatives that can be used as anticancer drugs [34]. The aryl analogs designed in this way feature better HDAC-inhibiting activity than the parent compound. These results suggest that the antiradical activity of AM can also be amplified by similar modifications using an aryl substituent with one (Figure 6), two, or three hydroxyl groups attached to the benzene ring. The results of the calculations for the simplest phenol–AM derivative (AMD, see Table 4 and Table 5) indicate that both the O-H and S-H moieties show greater antiradical activity in a vacuum and in an aqueous medium than the same groups activated separately in the parent molecules. The scavenging mechanism predicted for AM and AMD in a vacuum for active O-H and S-H groups is HAT. In the case of the S-H group, BDE(AM) = 86.27, whereas BDE(AMD) = 85.51 [kcal mol^−1^]; hence, the influence of the aryl substituent on the S-H activity is marginal. In the case of the O-H group, BDE(phenol) = 85.54 and BDE(AMD) = 81.33 [kcal mol^−1^], so the AM constituent significantly affects the activity of the O-H moiety due to the conjugation of the bonds of the benzene ring and the propene chain, which weakens the O-H bond. In water, SPLET is the preferred mechanism for both moieties. As the values of PA(AM) = 49.85 and PA(AMD) = 50.09 [kcal mol^−1^] are for S-H almost equal, the effect of the aryl substituent on the S-H reactivity is negligible in the first SPLET step. The influence of this type is observed in the second stage of SPLET owing to ETE(AM) = 83.18 and ETE(AMD) = 79.44 [kcal mol^−1^]. In the case of the O-H group, PA(phenol) = 52.71, PA(AMD) = 50.19, ETE(phenol) = 77.65, and ETE(AMD) = 74.45 [kcal mol^−1^], which indicates that the AM constituent significantly affects the antiradical activity of the O-H moiety in AMD at the first and the second stages of SPLET. Due to the strong antiradical and antimicrobial activity of both phenol and AM, their combination could be a valuable ingredient in cosmetics, food, and supplements, with AMD playing a dual role as a preservative and a radical scavenger. Assuming that changing the aryl group to the phenolic one retains the anticancer activity of AMD, this derivative seems to be a candidate for a component in sunscreens, which protect against the carcinogenic influence of UVA and UVB rays on skin and the production of harmful teratogenic radicals. The results of the calculations performed for the simplest phenol–AM system are promising to such an extent that they suggest extending the research to a hypothetical benzenetriol–AM system, which would combine the strong antiradical and antimicrobial properties of the two parent compounds. Investigations of selected benzenetriols (pyrogallol, hydroxyquinol, and phloroglucinol) are in progress.

## 4. Conclusions

A comparison of the BDE values evaluated for selected well-known antioxidants with the value of BDE = 86 [kcal mol^−1^] obtained for AM indicates that the potency of this compound to eliminate free radicals is identical to that characterizing α-lipoic acid and its metabolites, comparable to that of trans-p-coumaric acid, and greater than that of trans-ferulic acid. We conclude that although AM is a weaker radical scavenger than PSA, it nevertheless exhibits potent antiradical activity. Therefore, AM can be used not only as a flavoring food additive but also as a radical scavenger, protecting food, cosmetics, and drugs against the physicochemical degradation caused by reactive radical species. It should be emphasized that the highest concentration of AM is found in crushed garlic, whereas only traces are detected in the aged ethanol extract [35]. Consequently, the optimal way to utilize its antiradical properties is to use freshly processed garlic cloves. Recent studies have revealed the presence of the new radical scavengers in garlic macerates (ajothiolanes) [36] and in extracts of garlic skins (peels) [37] (N-trans-coumaroyloctopamine, N-trans-feruloyl-octopamine, guaiacyl-glycerol-β-ferulic acid ether, and guaiacyl-glycerol-β-caffeic acid ether). Therefore, other active compounds besides those considered here may also be responsible for the direct antiradical activity of garlic in processed forms. In view of this, it can be argued that the total antiradical activity of garlic is caused by a complex of compounds that scavenge free radicals both directly and indirectly as a result of the activation of endogenous antioxidants and other radical deactivation pathways mentioned in this work. The type of compound acting directly depends on the form of garlic processing, and, in the case of freshly crushed cloves, it includes the short-acting PSA and the long-acting AM.

## Data Availability

All new data created are reported in this work.

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
