# Peer review of "Density Functional Theory Studies on the Chemical Reactivity of Allyl Mercaptan and Its Derivatives"

_molecules, 2024, doi:10.3390/molecules29030668_

Round 1

Reviewer 1 Report

Comments and Suggestions for Authors

The manuscript is recommended for acceptance pending clarification on the following questions.

  1. How was the DFT method employed in the study, and why was the B3LYP/cc-pVQZ theory level chosen for the calculations?
  2. What role does the C-PCM solvation model play in the comparative analysis of the reactivity of garlic metabolites 2-propenesulfenic acid (PSA) and allyl mercaptan (AM)?
  3. Can you elaborate on the significance of the thermodynamic descriptors (BDE, PA, ETE, AIP, PDE, Gacidity) in determining the reactivity of PSA and AM?
  4. What are the global descriptors of chemical activity, and how do ionization potential (IP), electron affinity (EA), chemical potential (μ), absolute electronegativity (χ), molecular hardness (η), and softness (S) contribute to the understanding of the reactivity of the studied compounds?
  5. How do the calculated results indicate that PSA is more reactive than AM, and what implications does this have for the potential applications of these compounds?
  6. Could you explain the observed role of allyl mercaptan (AM) in deactivating free radicals and its greater chemical stability compared to PSA?
  7. In what ways does the presence of a double bond in AM contribute to its polymerization and preservation of antiradical activity, especially in the context of the S-H group?
  8. How does the aryl substituent containing hydroxyl groups affect the antiradical activity of AM and its derivatives, as indicated by the calculations?
  9. Can you provide insights into the antiradical activity of the simplest phenol-AM derivative, and how do both O-H and S-H moieties exhibit greater antiradical activity in different environments (vacuum and aqueous medium)?
  10. Based on the results obtained, how might allyl mercaptan (AM) and its derivatives find applications beyond being flavoring food additives, particularly in protecting food, supplements, cosmetics, and drug ingredients against physicochemical decomposition caused by exogenous radicals?

Comments on the Quality of English Language

The manuscript is recommended for acceptance pending clarification on the following questions.

  1. How was the DFT method employed in the study, and why was the B3LYP/cc-pVQZ theory level chosen for the calculations?
  2. What role does the C-PCM solvation model play in the comparative analysis of the reactivity of garlic metabolites 2-propenesulfenic acid (PSA) and allyl mercaptan (AM)?
  3. Can you elaborate on the significance of the thermodynamic descriptors (BDE, PA, ETE, AIP, PDE, Gacidity) in determining the reactivity of PSA and AM?
  4. What are the global descriptors of chemical activity, and how do ionization potential (IP), electron affinity (EA), chemical potential (μ), absolute electronegativity (χ), molecular hardness (η), and softness (S) contribute to the understanding of the reactivity of the studied compounds?
  5. How do the calculated results indicate that PSA is more reactive than AM, and what implications does this have for the potential applications of these compounds?
  6. Could you explain the observed role of allyl mercaptan (AM) in deactivating free radicals and its greater chemical stability compared to PSA?
  7. In what ways does the presence of a double bond in AM contribute to its polymerization and preservation of antiradical activity, especially in the context of the S-H group?
  8. How does the aryl substituent containing hydroxyl groups affect the antiradical activity of AM and its derivatives, as indicated by the calculations?
  9. Can you provide insights into the antiradical activity of the simplest phenol-AM derivative, and how do both O-H and S-H moieties exhibit greater antiradical activity in different environments (vacuum and aqueous medium)?
  10. Based on the results obtained, how might allyl mercaptan (AM) and its derivatives find applications beyond being flavoring food additives, particularly in protecting food, supplements, cosmetics, and drug ingredients against physicochemical decomposition caused by exogenous radicals?

Author Response

Reply to Reviewer I

Answers (A) to Comments and Suggestions for Author (Q)

Q1. How was the DFT method employed in the study, and why was the B3LYP/cc-pVQZ theory level chosen for the calculations?

A1. The DFT method was employed to calculate the total energy of neutral, radical, cation and anion forms of PSA and AM as well as its derivatives, indispensable in determining the chemical and thermodynamic parameters characterizing the reactivity of the compounds considered. The test calculations performed for PSA in water and the C-PCM solvation model revealed that the B3LYP theory level yields the smallest value of the total energy for this molecule in comparison to the energies generated by other functionals (e.g. M06–2X, ωB97XD, BHandHLYP, CAM-B3LYP). For example: E(M06-2X) = -591.307805, E(ωB97XD) = -591.347034, E(BHandHLYP) =-591.294711, E(CAM-B3LYP) = -591.330316 [Ha] in comparison to E(B3LYP)=-591.436087 [Ha]. The basis set cc-pVQZ used in the calculations is a compromise between the expensive calculation time and the accuracy of the generated results. Application of the more advanced basis set aug-cc-pVQZ showed that the difference in the calculated PSA energy for the above basis sets differs by 1.1 [kcal/mol] - the accuracy required in realistic chemical predictions.

Q2. What role does the C-PCM solvation model play in the comparative analysis of the reactivity of garlic metabolites 2-propenesulfenic acid (PSA) and allyl mercaptan (AM)?

A2. In a comparative analysis, it is necessary to use the same calculation methodology for all descriptors and molecules investigated. The choice of the C-PCM model is a consequence of the test calculations performed for gallic acid (M. Molski, Heliyon 2023, 9, e12806), which revealed that the combination of the B3LYP method and the C-PCM solvation model correctly reproduces the difference in HOMO - LUMO energies. Since this difference determines the values of chemical activity parameters, the C-PCM model was employed in the calculation of both chemical activity and thermodynamic descriptors for molecules under consideration.

Q3. Can you elaborate on the significance of the thermodynamic descriptors (BDE, PA, ETE, AIP, PDE, Gacidity) in determining the reactivity of PSA and AM?

A3. The thermodynamic parameters mentioned describe the energy needed to activate a given radical scavenging mechanism (HAT, SPLET, SET-PT, TMC) associated with the descriptor. For this reason, using the minimum energy criterion, it is possible to indicate which mechanism of radical deactivation is preferred by the molecule. On the other hand, analyzing descriptors calculated for different molecules, one may compare their anti-radical activity and indicate a more potent deactivator. In the case of Gacidity, a lower value of this parameter indicateing a greater ability to chelate the transition metal ions (especially Fe+2 and Cu+2), which produce stable complexes, remove them from the reaction medium, and slow down radical processes.

Q4. What are the global descriptors of chemical activity, and how do ionization potential (IP), electron affinity (EA), chemical potential (μ), absolute electronegativity (χ), molecular hardness (η), and softness (S) contribute to the understanding of the reactivity of the studied compounds?

A4.  The relationship between chemical reactivity and the type of descriptor determined is described in the Supplementary Information attached to the manuscript. For example, a large difference between HOMO-LUMO energies characterizes a hard molecule that is more stable and less active, while a small energy gap defines a soft molecule that is less stable and more reactive. The higher value of the electrophilicity index ω implies that the compound can be considered a strong electrophile, hence a strong nucleophile is described by lower values of ω. This parameter can also be related to a toxicity of a compound (M. Molski, Toxicol. Lett. 349 (2021) 30-39).

Q5. How do the calculated results indicate that PSA is more reactive than AM, and what implications does this have for the potential applications of these compounds?

A5. A comparison of the values of the descriptors determined: ∆E(PSA)=5.6461 [eV] vs ∆E(AM)=6.3773 [eV], (PA+ETE)(PSA)=114 [kcal/mol] vs (PA+ETE)(AM)=133 [kcal/mol] clearly indicates that PSA in water is a stronger radical deactivator than AM. The scavenging mechanism predicted for PSA and AM in the water medium is sequential proton loss electron transfer (SPLET) as the values of PA=58 and 50 [kcal mol-1] are less than BDE=68 and 87 [kcal/mol] as well as AIP=114 and 125 [kcal/mol], respectively. Additionally, ETE= 56 and 83 [kcal/mol], hence the second step of the SPLET mechanism also requires less energy than that indispensable to initiate the HAT or SET-PT antiradical processes. On the other hand, PSA is a short-lived species (less than 1 [s] in the gas phase). Therefore, the interval of its antiradical activity is strongly restricted, so it is impossible to use PSA as an active component of supplements, cosmetics or drugs in contradistinction to the more stable and long-lived AM. However, both substances may contribute to the antiradical activity of freshly crushed and directly consumed garlic cloves.

Q6. Could you explain the observed role of allyl mercaptan (AM) in deactivating free radicals and its greater chemical stability compared to PSA?

A6. The difference in PSA and AM reactivity (and stability) is a consequence of the type of chemical bonds -S-O-H and -C-S-H resulting in the BDE (Bond Dissociation Enthalpy) descriptor values: BDE(PSA) =68.04 [kcal/mol] and BDE(AM)=86.94 [ kcal/mol] in water. Those values indicate that the O-H bond in PSA is more susceptible to breakdown than the S-H bond in AM. On the other hand, comparing BDE(AM)=86.27 [kcal/mol] (vacuum) with the values of BDE(GA) = 80 (3'O-H), 79 (4'O-H), 87 (5'O-H) [kcal/mol] calculated for the three O-H phenolic groups in GA (gallic acid - strong radical deactivator), we see that susceptibility of AM to hydrogen atom transfer is comparable to that predicted for GA. Consequently, AM can be viewed as a potent radical deactivator.

Q7. In what ways does the presence of a double bond in AM contribute to its polymerization and preservation of antiradical activity, especially in the context of the S-H group?

A7. The presence of a double bond in AM contributes to its polymerization in the same manner thaat propene (propylene) does to polypropylene created by breaking a double bond and linking monomers in the radical form. The double bond C=C in  the monomer and the single bond C-C in the polymer have a marginal effect on the antiradical activity of the S-H group in AM and its polymers.

Q8. How does the aryl substituent containing hydroxyl groups affect the antiradical activity of AM and its derivatives, as indicated by the calculations?

A8. The scavenging mechanism predicted for AM and the phenol-AM derivative (AMD) in vacuum for active O-H and S-H groups is HAT. In the case of the S-H group, BDE(AM)=86.27, whereas BDE(AMD)=85.51 [kcal/mol], hence the influence of the aryl substituent on the S-H activity is marginal. In the case of O-H group, BDE(Phenol)=85.54 and BDE(AMD)=81.33 [kcal/mol], so the AM constituent significantly affects the activity of the O-H moiety due to the conjugation of the bonds of the benzene ring and the propene chain, which weakens the O-H bond. In water, the mechanism preferred is SPLET for both moieties. As for S-H the values of PA(AM)=49.85 and PA(AMD)=50.09 [kcal/mol] are almost equal, the effect of the aryl substituent on the S-H reactivity is negligible in the first SPLET step. The influence of this type is observed in the second stage of SPLET because ETE(AM)=83.18 and ETE(AMD)=79.44 [kcal/mol]. In the case of the O-H group, PA(Phenol)=52.71, PA(AMD)=50.19, ETE(Phenol) = 77.65 and ETE(AMD)=74.45 [kcal/mol], indicating that the AM constituent significantly affects the antiradical activity of the O-H moiety in AMD at first and second stages of SPLET.  

Q9. Can you provide insights into the antiradical activity of the simplest phenol-AM derivative, and how do both O-H and S-H moieties exhibit greater antiradical activity in different environments (vacuum and aqueous medium)?

A9. The values of the two-stages descriptors for the phenol-AM derivative in water (AMD) (PA+ETE)(S-H)=400 and (PA+ETE)(O-H)=396 [kcal/mol] decrease to (PA+ETE)(S-H)=130 and (PA+ETE)(O-H)=125 [kcal/mol]. Those significant changes indicate that the SPLET scenario for AMD  is supported by the ionic interactions of O-H and S-H groups with polar water facilitating the AMD deprotonation in the first SPLET stage.

Q10. Based on the results obtained, how might allyl mercaptan (AM) and its derivatives find applications beyond being flavoring food additives, particularly in protecting food, supplements, cosmetics, and drug ingredients against physicochemical decomposition caused by exogenous radicals?

A10. Antiradical and antimicrobial properties of AM polymers can be employed in food packaging and storage, including encapsulation in AM biopolymers and covering food with microfilm containing active S-H groups. Assuming that changing the aryl group to the phenolic one retains anticancer activity of AMD, this compound seems to be a potential component of sunscreen creams that protect against the carcinogenic influence of UVA and UVB rays on skin and production of harmful teratogenic radicals.

Reviewer 2 Report

Comments and Suggestions for Authors

Comments:

The study employed DFT at the B3LYP/cc-pVQZ level and C-PCM solvation model to compare the reactivity of garlic metabolites, 2-propenesulfenic acid (PSA) and allyl mercaptan (AM). Thermodynamic and global descriptors were determined, revealing that PSA is more reactive, while AM, with its greater stability and longer lifetime, may deactivate free radicals. AM's double bond allows polymerization, preserving its antiradical activity. Derivatives with aryl substituents show enhanced antiradical activity. Despite being a weaker antioxidant than PSA, AM exhibits potent antiradical activity, making it valuable for protecting food, cosmetics, and drugs against degradation by reactive oxygen species. Freshly processed garlic cloves are recommended for optimal utilization of AM's antiradical properties. The total antioxidant activity of garlic involves a complex of compounds acting directly and indirectly, with PSA and AM playing key roles in certain processed forms. The study highlights the importance of considering various compounds in understanding garlic's antioxidant activity. The manuscript is well-organized and clearly stated. I would suggest accepting it after the following concerns are addressed. 

Major:

The entire “Introduction” part is divided into 2 paragraphs. The second paragraph is too lengthy, from lines 55-130. Here are some suggestions for this introduction:

·       The logical connection between paragraphs can be smoother and more natural.

1.     The author wrote in line 55-57 that “Then it undergoes self-condensation to convert allicin to diallyl disulfide and AM, volatile compounds responsible for a specific odor of crushed garlic and, presumably, its health benefits.” The object referred to by "it" here is not clear, which can easily make readers confused. Avoiding the use of pronouns that can easily cause ambiguity is something that should be paid attention to in academic writing. Clear reference can enhance the logic and readability of the article.

·       The scientific and innovative nature of the research content is not outstanding enough, and the current version is not clear enough in this area.

·       The background introduction in the preface is long, and the sentences can be streamlined to highlight key information. The literature review can focus on literature directly related to the research question and avoid citing too much background knowledge.

·       Some statements are not concise and logical enough and could be further refined to emphasize the core points.

1.  The author wrote in line 33-35 that “Among the properties mentioned, the antioxidant activity [2-5] of garlic deserves special attention, as the mechanism of neutralization of reactive free radicals is not yet fully recognized.” The various bioactive properties of garlic, particularly its antioxidant activity, deserve special attention, which is reasonable. However, the subsequent reason provided, "the mechanism for neutralizing reactive oxygen species is not fully understood," does not establish a strong logical connection between the antioxidant activity and the point worth noting.

I hope that the revised introduction will have a clearer overall organization, a stronger sense of hierarchy, and more concise and refined sentences.

​

In this manuscript, it is recommended to split the longer sentence into two shorter sentences to improve readability and comprehension.

1.     For example, in 108-203, one  sentence is that “Additionally, values of ω-=4.5746 (4.2353) indicate that the antioxidant activity of PSA is weaker than that assigned [22] to β-carotene, astaxanthin, psittacofulvins (pigments found in parrot feathers) and stronger than vitamin A, C, E, and anthocyanins, while the antioxidant power of AM is greater than that reported [22] for vitamin A, E, anthocyanins (except delphinidin) and smaller than the remaining cases under comparison [22].

Minor :

1.     In line 128,it is recommended to use "Fe2+ and Cu2+" instead of "Fe+2 and Cu+2".

2.     In Scheme 1, SOH has two areas that need improvement.

3.     In Scheme 2,ROOH and arrows overlap somewhat.

Comments on the Quality of English Language

English in this manuscript is good.

Author Response

Reply to Reviewer II

Thank you for your constructive review of the manuscript. Following your suggestions and comments, I made significant changes to the manuscript to include your valid objections. In particular:

  1. The chapter Introduction has been divided into two parts: I. Introduction and II. Computation details.
  2. Too long sentences have been split into two shorter ones to improve readability and comprehension.
  3. In case of ambiguity of the sentences, the pronouns "it" and "they" have been replaced with the name of the objects referred.
  4. The notation Fe2+ and Cu2+instead of Fe+2 and Cu+2 has been used.
  5. In Scheme 1 two SOH areas have been improved, whereas in Scheme 2 the overlapping symbols have been removed.

The revised version of the manuscript has a much clearer overall organization and contains more concise and refined sentences.

Round 2

Reviewer 2 Report

Comments and Suggestions for Authors

I suggest it can be accepted in its current form.